# New Ways of Working through Emerging Technologies: A Meta-Synthesis of the Adoption of Blockchain in the Accountancy Domain

Rocco Agrifoglio [1] and Davide de Gennaro [2,*]

1   Department of Business and Economics, University of Naples "Parthenope", 80132 Naples, Italy; rocco.agrifoglio@uniparthenope.it
2   Department of Business Sciences—Management & Innovation Systems, University of Salerno, 84084 Fisciano, Italy
*   Correspondence: ddegennaro@unisa.it; Tel.: +39-3341573120

**Abstract:** In an attempt to deepen how the way of working is changing due to the digital transformation, this research aims at understanding the process by which individuals adopt blockchain technology in accountancy. We conducted a meta-synthesis of the qualitative literature on the topic of blockchain technology adoption in the context of accountancy. Drawing from 10 systematically selected qualitative studies, we analyzed the process of blockchain technology adoption in accountancy, with particular reference to the impacts on accounting professionals, in terms of individual attitudes and behaviors, as well as organizations. Our findings contribute to the existing literature in at least two ways. First, our research explores the topic of blockchain adoption in the accountancy domain and stresses the relevance of the use of that emerging technology by accounting professionals and organizations, as well as the main problems that could limit its adoption and use. Second, we provide an overview of the process of blockchain technology adoption with specific reference to the questions of "why" and "how" blockchain is (or is not) adopted by accounting professionals and organizations, in an effort to shed light on a critical issue that has yet to be explored in accountancy.

**Keywords:** blockchain technology; accountancy; digital transformation; meta-synthesis; qualitative study; review

## 1. Introduction

Blockchain is a subfamily of technologies in which the ledger is structured as a chain of blocks containing transactions and whose validation is entrusted to a consensus mechanism [1]. The main features of blockchain technologies are registry immutability, transparency, transaction traceability, and security based on cryptographic techniques [2,3]. Business is based on information; the faster and more accurate, the better. Blockchain technology is ideal for transmitting this data because it provides immediate, shared, and fully transparent information stored in an immutable ledger that can be accessed only by authorized network members. A blockchain network can, among other things, track orders, payments, accounts, production, and more. Furthermore, because members share an unambiguous view of the truth, it is possible to see all the details of a transaction end-to-end, thus generating greater trust as well as new opportunities in terms of efficiency.

While there are different types of platforms using blockchain technology, each with their own configurations, they all have at least some points in common [4,5]: (i) the digitization and transformation of data into a digital format; (ii) decentralization, whereby images are distributed among multiple nodes to ensure the cybersecurity of the systems; (iii) the traceability of transfers, whereby each step is traceable in every part and its origin is recorded; (iv) disintermediation, whereby transactions are managed without intermediaries (i.e., without the intervention of trusted central entities, such as banks); (v) verifiability,

whereby every element of the register is transparent, visible to all, and therefore totally consultable and able to be verified; (vi) the immutability of the registry, whereby the registry data cannot be modified without the consent of the network (often referred to as net neutrality); and (vii) the programmability of the transfers, whereby it is possible to program some actions that are activated only when certain prearranged conditions are verified.

The applications of blockchain are relevant in a variety of industries and make it possible, at least potentially, to track assets in a business network, which can be tangible (houses, cars, money, land) or intangible (intellectual property, patents, copyrights, trademarks); indeed, virtually anything of value can be tracked and traded on a blockchain network, reducing risks and costs for all involved [6,7]. The relationship between technology and cryptocurrencies, for example, is critical to understanding the importance of blockchain [8].

Although blockchain technology has received considerable interest in some subject areas—with the most advanced being the finance and insurance sector, which was initially threatened by bitcoin, but also including agri-food, advertising, logistics, and even public administration—the same cannot be said with respect to the accounting, auditing, and accountancy fields [9]. Starting with the characteristics and definitions of blockchain in accountancy—which remain unclear [10,11]—few studies analyze which aspects of this technology can be applied to accounting [12]. Although there are interesting contributions to this topic, at the moment the accounting industry shows little theoretical propensity with reference to blockchain [13]. Another area in which it is important to continue to research is the human–algorithm duality perspectives in the auditing process [14]. Many research ideas could still be validated, since the topic still needs investigation [9,15]. Another gap in the blockchain literature concerns research on qualitative approaches [16]. Qualitative methodology allows for the expansion of knowledge on how blockchain can be used in supply-chain management but without losing sight of the context in which the study is conducted [17,18]. Moreover, it is important to analyze qualitative studies on the topic, because blockchain is still in its infancy and needs further study on the processes underlying this phenomenon [19]. Therefore, it is critical to promote new empirical research, especially using a qualitative approach, to create a virtuous collaboration between academics and practitioners [9].

Our research aims at understanding the process by which individuals adopt blockchain technology in accountancy. Using a meta-synthesis of qualitative research, we selected 10 qualitative studies with the aim of exploring the process of blockchain technology adoption in accountancy by focusing on the impacts on accounting professionals and organizations.

Our research contributes to the existing literature in at least two ways. First, our research explores the topic of blockchain adoption in the accountancy domain and stresses the relevance of the use of this emerging technology by accounting professionals and organizations, as well as the main problems that could limit its adoption and usage. Second, we provide an overview of the process of blockchain technology adoption in light of the questions of "why" and "how" blockchain is (or is not) adopted by accounting professionals and organizations, so as to shed light on a critical issue that remains to be explored in accountancy. Theoretical and practical implications for future research will be offered.

## 2. Theoretical Framework

The concepts of bitcoin and blockchain technology were introduced by an individual or group with the pseudonym Satoshi Nakamoto, who penned the original Bitcoin Whitepaper [1]. Nakamoto provided a detailed description of a digital currency, namely Bitcoin, by highlighting how its application can solve the double-spending problem using cryptology proof and open distributed ledgers. In this regard, blockchain was the technology underlying Bitcoin that enables the existence and running of such cryptocurrency. From 2008 to today, based on the work of Nakamoto and other developers, the evolution of blockchain has been a progressive process making it possible to extend the range of

applications beyond finance, thereby attracting increasing attention from many scholars and practitioners from various fields and countries [20,21].

Although originally conceived as the basis of cryptocurrencies, blockchain is now recognized as the technology that makes it possible to carry out transactions in a permanent, distributed, and digital ledger [22]. Under the blockchain technology, users are called nodes, and the underlying idea is the existence of a peer-to-peer (P2P) network which enables the verification and storage of electronic transactions, as well as the integral reproduction of the digital archive in all the nodes of a network. Transactions are recorded in the ledger with a digital signature based on public key cryptography, also known as asymmetric cryptography [20]. The certification of transactions is ensured by the blockchain users, such that each transaction is validated by the nodes of the network using a consensus mechanism (or protocol). Therefore, the blockchain technology enables a network in which the various parties can interact even without trusting each other, as it is the network that acts as a guarantor through the mechanism of consent by the nodes for any type of operation [23].

Under an organizational perspective, the adoption of blockchain technologies encourages the development of external relations and inter-firm networks, thus enabling the creation of new business models aimed at increasing collaboration to encourage synergy, innovation, and economic development. The literature agrees that blockchain and distributed ledger technologies enhance trust in business relations and allow for the improvement of existing relations and the creation of new inter-firm networks [24]. In this way, the trust and confidence arising from distributed ledger technologies has enabled the application of blockchain in multiple domains, such as finance, banking, trading, healthcare, public administration, agri-food, assurance, and accounting.

A growing number of authors have recently focused on the application of blockchain technologies in the accountancy domain [12,25–29]. For instance, blockchain was recognized as a particularly suitable technology for achieving more real-time aggregation and for reducing the time consumed in highly standardized processes (i.e., reconciliation of accounts, observation and inquiries, inspection of records, etc.), as well as for the sharing of practitioner misconduct [12,27]. Blockchain also allows organizations to reduce costs and human errors by automating transactions through smart contracts, as well as to avoid manipulation and fraud thanks to instant sharing of information and enhancing information integrity [12,28]. A "smart contract" is defined as a computer program that operates on distributed ledger-based technologies and whose execution automatically binds two or more parties based on effects predefined by them. By being decentralized transparent, and consensus protocol-based, blockchain technologies enable a permanent and immutable record of financial transactions, thus increasing trust and auditability of accounting information [30]. Tiron-Tudor and colleagues [29] provided a systematic literature review on how accounting organizations might manage the changes induced by blockchain technology advancement. Since the accountancy organizations possess typically bureaucratic and rigid structures due to being based on a large number of repetitive tasks and operations, the usage of blockchain in accounting might be affected by individuals' beliefs, in terms of performance and social influence, as well as management's capabilities to manage organizational change and challenges related to emerging technology usage.

However, despite this proliferation of contributions, qualitative studies are often undervalued and not included when referring to the blockchain [16]. In contrast, it is notable that, with some exceptions, earlier studies were primarily qualitative, and, similar to the quantitative approach, this qualitative literature continues to grow. Each qualitative study is unique [31]; as a result, the qualitative literature is more varied from the quantitative literature thus requires further investigation [9,19]. For example, qualitative studies may focus on the emotional and behavioral impact of blockchain technology on workers, while quantitative studies may focus on more objective, measurable variables related to specific forms of technology adoption. In practice, different approaches seek to answer different research questions, whereby qualitative studies focus on understanding the reasons, opinions,

and motivations of individuals, whereas quantitative studies focus on a priori identified variables to be tested empirically [32].

Moreover, it is still unclear what the real scope of artificial intelligence and blockchain technology might be [33] as well as what their potential is with respect to the accounting, auditing, and accountability fields [9–12,15], with which there seems to be little theoretical fit [13].

Thus, to address these gaps in the literature while responding to calls from scholars [8,18], this study aims at expanding the knowledge about blockchain through a meta-synthesis of 10 qualitative studies analyzing the impact of the technology on individuals' behaviors in the accountancy industry.

## 3. Method

To understand and describe key and recurring insights and topics about blockchain technology in the accountancy literature, we conducted a meta-synthesis of qualitative evidence [34,35]. Similar to meta-analysis, a meta-synthesis integrates and combines studies on a specific topic in order to produce comprehensive and interpretive findings [36]. Through such an analysis, it is possible to understand the behaviors of individuals with respect to the topic of interest [37]. Our methodological approach was based on five sequential steps, represented graphically in Figure 1 and explained in detail in the following subsections, involving synthesizing qualitative arguments to create a process model.

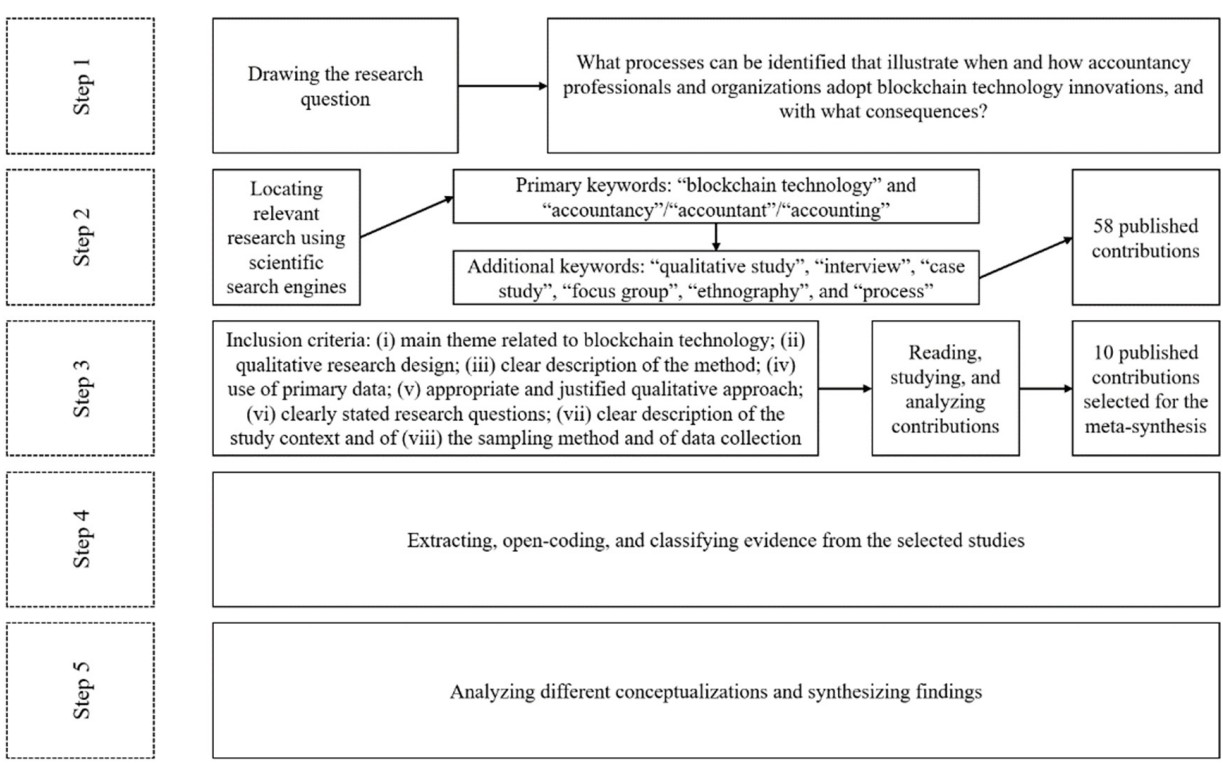

**Figure 1.** The five methodological steps of meta-synthesis.

### 3.1. Formulating the Research Question

Based on the reference literature on blockchain technology, we set up a meta-synthesis of the qualitative research that could provide insights into the process of acquiring technological innovation in organizations and the impact it has on workers. We therefore guided the meta-synthesis using the following research question: What processes can be identified that illustrate when and how accountancy professionals and organizations adopt blockchain technology innovations, and with what consequences?

### 3.2. Locating Relevant Research

The second step involved identifying studies that may be considered relevant to our meta-synthesis through a computerized search using scientific search engines (e.g., Scopus, Web of Science) and databases (e.g., EbscoHost Business Source Complete, Google Scholar) to find citations containing the primary keywords "blockchain technology" and "accountancy"/"accountant"/"accounting". Moreover, the search string included the following additional keywords: "qualitative study", "interview", "case study", "focus group", "ethnography", and "process". Finally, we also searched for published articles, conference proceedings, and book chapters using scientific networks, reference lists, and active authors in the field of blockchain to further identify consistent studies. Although a meta-synthesis should be comprehensive and include as many studies as possible [38], we planned to focus only on published studies. This choice provides scientific rigor, as peer-reviewed publication processes provide at least one benchmark to which to adhere when selecting articles [35].

### 3.3. Establishing Inclusion and Exclusion Criteria

We applied the following inclusion criteria to ensure that the sample of articles used for the analysis was appropriate: (i) the main theme of the study related to blockchain technology, and the study included (ii) a qualitative research design with data collected from focus groups, interviews, observations, or narrative approaches; (iii) a clear description of the method of analysis; (iv) use of primary data; (v) an appropriate and justified qualitative approach; (vi) clearly stated research questions; (vii) a clear description of the study context; and (viii) a clear description of the sampling method and of data collection. The initial search of published qualitative blockchain technology studies yielded a set of 58 published contributions, also obtained via cross-checking of reference lists, scientific networks, and active researchers in the field.

After a first screening of titles and abstracts, we found 16 studies to be false positives, since blockchain technology was not their main topic of investigation. After obtaining the studies' full-text versions, all authors independently reviewed all the papers and together reached a final agreement on the dataset for this study, excluding studies if they: were reviews or quantitative ($n = 23$); failed to state the research method, theoretical framework, or study context ($n = 3$); did not provide an adequate description of the research question ($n = 2$), the appropriateness of the qualitative methodology ($n = 1$), or the role of the researcher ($n = 2$); or did not use purposefully collected data (instead using, e.g., existing data that functioned as an illustrative example) ($n = 1$). After all the studies were read in their entirety, 10 of them met all the inclusion criteria and were ultimately incorporated into the meta-synthesis (Table 1).

**Table 1.** Details of studies selected for meta-synthesis.

| Authors | Year of Publication | Title | Journal/Book | Doi |
|---|---|---|---|---|
| Al-Htaybat, K., Hutaibat, K., von Alberti-Alhtaybat, L. [39] | 2019 | Global brain-reflective accounting practices: Forms of intellectual capital contributing to value creation and sustainable development | *Journal of Intellectual Capital* | 10.1108/JIC-01-2019-0016 |
| Boulianne, E., Fortin, M. [40] | 2020 | Risks and Benefits of Initial Coin Offerings: Evidence from impak Finance, a Regulated ICO | *Accounting Perspectives* | 10.1111/1911-3838.12243 |
| Cai, C.W. [41] | 2021 | Triple-entry accounting with blockchain: How far have we come? | *Accounting and Finance* | 10.1111/acfi.12556 |
| Dal Mas, F., Dicuonzo, G., Massaro, M., Dell'Atti, V. [42] | 2020 | Smart contracts to enable sustainable business models. A case study | *Management Decision* | 10.1108/MD-09-2019-1266 |

**Table 1.** *Cont.*

| Authors | Year of Publication | Title | Journal/Book | Doi |
|---|---|---|---|---|
| Helliar, C.V., Crawford, L., Rocca, L., Teodori, C., Veneziani, M. [43] | 2020 | Permissionless and permissioned blockchain diffusion | *International Journal of Information Management* | 10.1016/ j.ijinfomgt.2020.102136 |
| Massaro, M., Dal Mas, F., Chiappetta Jabbour, C.J., Bagnoli, C. [44] | 2020 | Crypto-economy and new sustainable business models: Reflections and projections using a case study analysis | *Corporate Social Responsibility and Environmental Management* | 10.1002/csr.1954 |
| Ozlanski, M.E., Negangard, E.M., Fay, R.G. [45] | 2020 | Kabbage: A fresh approach to understanding fundamental auditing concepts and the effects of disruptive technology | *Issues in Accounting Education* | 10.2308/issues-16-076tn |
| Roszkowska, P. [46] | 2020 | Fintech in financial reporting and audit for fraud prevention and safeguarding equity investments | *Journal of Accounting and Organizational Change* | 10.1108/JAOC-09-2019-0098 |
| Sandner, P., Lange, A., Schulden, P. [47] | 2020 | The role of the CFO of an industrial company: An analysis of the impact of blockchain technology | *Future Internet* | 10.3390/fi12080128 |
| Zheng, Y., Boh, W.F. [48] | 2021 | Value drivers of blockchain technology: A case study of blockchain-enabled online community | *Telematics and Informatics* | 10.1016/ j.tele.2021.101563 |

### 3.4. Extracting and Coding Data

The fourth step in the meta-synthesis approach was to extract, code, and classify evidence from the selected studies. We used an open-coding approach to classify insights generated by the researchers of the primary studies, with a particular focus on the "Results" and "Discussion" sections [35]. Initially, all the coauthors coded three articles together to have a shared protocol; next, they independently coded the other articles according to the defined criteria. Working with two coders reduces mistakes in data recording and avoids omitting relevant constructs.

We followed an iterative approach [49], continuously iterating between our data and the emerging conceptualizations and comparing codes by engaging in discussion when disagreements emerged. We used Cohen's kappa coefficient ($\kappa$) to estimate the level of agreement between the coders; this is a statistic used to assess qualitative item inter- and intra-rater reliability and is able to consider the potential of agreement occurring by coincidence.

Initially, when all the coauthors codified the first three studies, a fair level of agreement was reached: $\kappa = 0.27$. By discussing the reasons for disagreements, particularly regarding terminological differences, we were able to identify and correct differences in the encoding process. Following this phase, in the final coding process, the observed agreement—that is, the percentage of agreement between the judgments of two raters when they independently coded the same data—reached a value of $\kappa = 0.88$, reflecting excellent agreement between the raters [50].

### 3.5. Analyzing Different Conceptualizations and Synthesizing Findings

In the next step, we related the previously identified concepts and themes into theoretical categories so that the emerging constructs, based on the extant literature, were grounded in our data. In doing so, we lifted the data to a more theoretical level and moved from a case-specific to a cross-study level of analysis. Moreover, to obtain an outsider perspective and thereby verify our ideas, we engaged in informal (e.g., internal seminars with fellow department members) and formal (e.g., conference session) meetings with

other researchers not involved in the study to discuss emerging patterns and solicit critical questions regarding the data collection and analytical procedure.

Thus, after settling on a set of theoretical categories, we identified key aggregate theoretical dimensions. The last phase synthesized the concepts that had emerged to systematize findings related to qualitative accounting blockchain research and to formulate a process model. The findings are explained in detail in the following sections.

## 4. Results: A Process Model for Blockchain Adoption in Accountancy

Based on the selected studies, this meta-synthesis proposes a sequence of events in terms of what happens, in what context, and with what consequences when adopting blockchain technology in accounting. Figure 2 summarizes the model, which will be detailed in the following subsections.

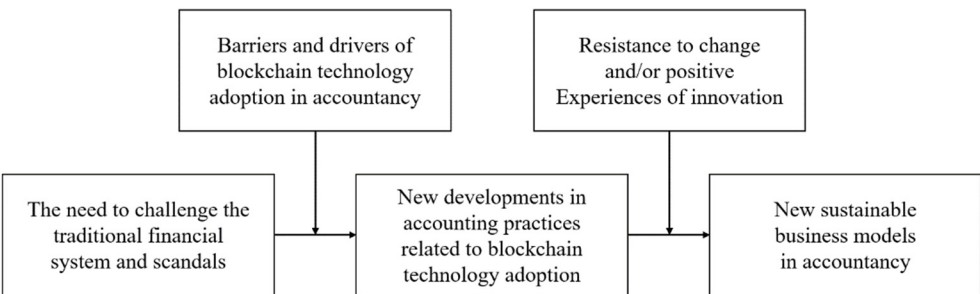

**Figure 2.** The process model for adopting blockchain technologies in accountancy.

### 4.1. The Need to Challenge the Traditional Financial System and Scandals

The starting point that emerged from the findings concerns a need to challenge and to overcome the traditional financial system and scandals. Although accountancy is one of the most important activities for organizations, especially for publicly traded companies, it took a failure of traditional auditing to start a discussion on how to regulate and improve this function [46]. Indeed, there are many cases of "creative accounting" and falsification of accounting records, whereby there is often a substantial discrepancy between the financial situation presented in the financial statements and the real one, resulting in an inability to estimate the actual profits and market value of companies.

The most notorious cases of fraudulent financial reporting and the inability of auditors to prevent it are the Enron and AA scandals [46]. These scandals stemmed from problems such as falsification of records and mutability of information; in some cases, certain loopholes in the law allow the true financial position to be misrepresented in the accounting records.

Another problem is caused by the diffusion of liability in organizations, which makes it very complicated to trace the identification of the authorship of individuals' actions, especially in large companies [51]. When many colleagues are present, a given person is less likely to take responsibility for an action or omission. Thus, all actions result from the aggregate actions of many individuals, making it difficult to assign proper responsibility. Similarly, accounting firms do not want to do anything to jeopardize their own income, so they are inclined to make the client happy and thus co-participate in misconduct and fraudulent practices [52].

Finally, another critical element lies in the fact that audit quality and auditor independence are two variables that should walk hand in hand [53]. When one is too dependent on a client, the ability to uncover and disclose any misstatements that could harm investors evaporates.

Major financial crises, such as the global economic crisis of 2007–2008 that started in the U.S. with the failure of high-risk (subprime) mortgage loans, have generated a distrust of financial institutions around the world [40]. In light of these problems and inefficiencies, the authors of the studies selected for this meta-synthesis identify a process suggesting

how blockchain, smart contracts, the Internet of Things (IoT), and machine learning (ML) can help safeguard equity investments in capital markets by improving the reliability of financial statements and improving the quality and outcomes of the audit process [46]. In fact, blockchain technologies were conceived early on as an antidote to the inequities and corruption of the traditional financial system.

### 4.2. Barriers and Drivers of Blockchain Technology Adoption in Accountancy

Blockchain technology is a tool with which to conduct transactions in a secure, immutable, and chronological manner without the need for intermediaries [54]. It is about digital information chains, distributed across multiple computers called nodes, located anywhere in the world. According to diffusion theory, a technological innovation such as blockchain goes through a few stages, including knowledge, persuasion, decision, implementation, and confirmation. Many studies in the literature have analyzed blockchain innovation without applying it to a particular industry or business sector, while Helliar and colleagues [43] have tried to contextualize it to the accountancy and financial sector.

The first application of blockchain dates back to 2009 and involved the cryptocurrency Bitcoin. Although the potential of blockchain was recognized early on by various disciplines, the literature has highlighted a number of barriers to adoption in the initial lack of knowledge and expertise regarding the innovation [43]. Blockchain ecosystems require many participants, not just technical experts, and this implies the necessity of the education and participation of many stakeholders to enable proper deployment in accountancy. Other barriers include issues of scalability, due to the low number of transactions that can potentially be processed [55], and interoperability and compatibility with other systems [56], as well as the more general costs of implementing new technologies [57]. Scalability is important since there are far fewer transactions per second than in other financial processing systems [58]. Computing power and electricity are other notable barriers [57], with many kilowatt hours being required to solve mathematical algorithms. Finally, changing regulations and the lack of appropriate governance are other barriers to blockchain deployment in accountancy [59], although many jurisdictions are making legal changes to support blockchain development, and this barrier can thus potentially become a driver for its adoption.

In addition to these barriers, there are also drivers of blockchain deployment [43]. Examples include transactional efficiency and disintermediation [60] as well as data immutability and integrity [55,57] and protection against hacking. In fact, early blockchain adoption cases were characterized by a reduction in intermediation costs, enabling blockchains to increasingly become clearinghouses for securities and financial transactions that can replace the multitude of actors involved in traditional financial processes. With reference to the immutability and integrity of data, this assurance could represent significant cost savings [61]. In addition, ring signatures and private keys ensure the privacy of the data of the people involved [55]. Blockchains also facilitate the Internet of Things (IoT) and Industry 4.0, which require thousands of devices to be connected together to process transactions and industrial processes, thus improving productivity and efficiency [57].

Overall, there are a number of barriers and diffusion factors that may hinder or spur the adoption of blockchain technology in accountancy. In this study, and based on the reviewed studies, we focus primarily on successful adoption in order to gain insight into the techniques used in this field and understand the underlying process.

### 4.3. New Developments in Accounting Practices Related to Blockchain Technology Adoption

Technology has profoundly changed the ways of life and work, impacting the private and relational sphere of all individuals. New developments in blockchain technologies have indeed ensured the emergence of new business models and practices by disrupting existing models; delivering instant, intimate, frictionless value on a large scale; and focusing more on delivering value for customers and other stakeholders. This is especially the case in accounting [39,45]. These new business models are based on agility in connecting anyone

and anything anywhere, as well as responding to unforeseen and unexpected external circumstances [62]. All actions are increasingly interdependent on each other, as illustrated by the concept of the global brain [39]. This is a metaphor for the total connectivity that is developing in the world and a reflection of evolutionary cybernetics in accounting [63]. These changes allow people to connect with each other and be part of the system in an unregulated way.

Accounting and intellectual capital practices in the context of new digital technologies [64] involve the combination of human and mechanical skills and the subsequent generation of value for organizations and society at large. Accounting practices can be used to relate certain images and narratives so that information can be visualized via augmented reality [39]. Incorporating augmented reality into accounting in order to visualize and synthesize capital flows and business model developments for value creation represents a significant advancement of digital technology in this area.

Through machine-to-machine communication, and thanks to intelligent and interconnected technological components, all the data exchange between nodes is automated without the need for human intervention. In fact, blockchain technology is the key to facilitating the complete integration of data flow of all the functions involved in an economic process. Importantly, the blockchain also allows for the automation of a firm's accounting and payment processes through the potential integration of machines into payment networks by using digital representations of conventional currencies [47].

The blockchain in accounting assumes a concept called "triple-entry accounting" [41]. This theory, which originated in the 1980s, suggests that, in addition to debit and credit entries, a third layer called "trebit" should be added to provide mode momentum financial information to the organization, enabling better strategic decision making. According to Cai [41], in a blockchain ecosystem, for some accounts, business entities will only need to perform a single entry internally, and the opposite entry will be recorded in a publicly shared ledger; moreover, this triple-entry accounting represents a new and more efficient way to address fundamental trust and transparency issues that plague current accounting systems.

### 4.4. Resistance to Change and/or Positive Experiences of Innovation

Although the benefits of these technological implementations in the way of working and organizing work are evident, there are still many examples of resistance to change in the accounting industry [41]. On the one hand, accounting professionals and academic researchers do not have adequate training on blockchain concepts and infrastructure and therefore do not possess sufficient knowledge and skills for effective use; on the other hand, to transform business processes (including accounting), blockchain experts need more support from the accounting profession and academia in terms of specific business and accounting knowledge [41]. This is undoubtedly a new way of interpreting the profession: for example, communication between client and accountant has gradually become increasingly virtual, and technology has made it possible to do practically anything at a distance. One example of this can be seen in digital signatures, which today are considered perfectly legitimate to validate documents but until very recently were either not even contemplated or were considered suspect [39].

Accounting firms have therefore had to become more efficient, and they have turned to tools that can help them to do so [48]. These include not just "simple" digital media (e.g., emails and calendars) but also extremely complex integrated software that can perform different tasks while saving time and greatly decreasing margins of error. Such software can be used by both professionals and their clients to keep everything under control, as it can be accessed from different devices in different locations. Moreover, immediate communication tools [45], such as chat applications, are more "responsive" than email and even phone calls. However, doubts remain in some cases about the use of software, especially regarding the auditing process [44]. One of the biggest doubts lies in the perplexity regarding the real advantage of using technology [41], when on the contrary, an artificial intelligence module

can support the professional in the most difficult steps, optimizing work and freeing up time to acquire more assignments.

Digitization has opened the door to new competitors in addition to those that naturally and historically preside over some of the accounting profession's own activities and are, in fact, now perceived as real threats [43]. Businesses' service innovations and client management are still little evolved, demonstrating that digital orientation is still characteristic of only a few [40]. It is necessary to create the conditions for digitization and artificial intelligence to create real opportunities for accountants and to enable accountants to seize those opportunities. Today, the focus is more on the concern that automation will, in the near future, replace the professional in performing low value-added activities than on the strategic management of this phenomenon [41].

Thus, a number of new opportunities for client service emerge, such as assistance in digitizing the tax function, or the outsourcing of compliance and ongoing management activities through the use of the most advanced technology solutions. These represent a good way to reduce operating costs, create efficiencies, and increase the effectiveness of a service delivery in a competitive environment, requiring a transformation strategy to take advantage of the opportunities offered by digital technology.

### 4.5. New Sustainable Business Models in Accountancy

The evolution of accounting has passed through various eras, and each evolution has been followed by a phase of accounting adaptation. Computer systems, in the last century, have led to integrated ways of recording, and blockchain is also set to impact these systems [41]. Technological progress undermines the paradigms of trust, and it is at this point that the blockchain intervenes, trying to restore trust by elevating it to the top of technology [48].

A blockchain system for recording economic and financial transactions can lead to profound changes in the way accounting and financial managers operate. Keeping records on a blockchain system would, in theory, lead to the elimination of "earnings management" risk in financial reports, as all financial events would be recorded on the blockchain as they occur [47]. The blockchain would enable the adoption of common standard guiding principles, the reduction of total reporting costs, and the description of the relevance of issues with a certain reliability and especially comparability of information. Indeed, blockchain technology in accounting fosters insight into the strategic goals of the organization; its ability to generate and maintain value in the short, medium, and long term; and the capital and relationships on which it depends [40].

However, uncertainty is not a sufficient reason to exclude the reporting of such information from final trust data. Furthermore, blockchain technology enables and emphasizes connections between different components of the business model, external factors that impact the organization, and the capital and relationships on which it depends [42].

Moreover, blockchain technology ensures a description of an organization's relationships with its key stakeholders: through blockchain technology, relevant information can be distinguished accountably in the market dimension [43]. In addition, blockchain extracts the concise and targeted information that enables the assessment of an organization's ability to create and maintain value in the short, medium, and long term. Everything is findable, complete, reliable and is based on trust. All information is findable and certified. In short, blockchain technology makes one imagine reporting comprising less paper but more substance [43,44].

From the studied articles, some new tools that are both related to blockchain and applicable to the context of accountancy emerge, since blockchain is one of the most interesting of the emerging smart technologies that allow companies to show and create knowledge and increase their competitiveness [44,65]. Massaro and colleagues [44] state that blockchain technology enables a secure and convenient business wherein customers can save money by acting as a community of committed people who are willing to cooperate and gain mutual benefits by supporting each other. The approval mechanism allows the

franchise and risk to be divided among multiple people, making the transaction more financially sustainable by increasing the ratio of resources used to output obtained.

As stated by Dal Mas and colleagues [42], blockchain-based platforms and smart contracts can replace centralized platforms, since these reduce transaction costs that would otherwise result from opportunism and uncertainty. In fact, smart contracts allow for the automatic self-enforcement of a contract under predetermined conditions. Therefore, this new technology reduces the costs of gathering and processing information, drafting and negotiating contracts, monitoring and enforcing agreements, and managing relationships, allowing for more market-based governance structures [66]. Automated decision making facilitated by smart contracts can reduce the impact of bounded rationality on transactions, therefore reducing uncertainty [67].

Interestingly, due to its unique features, blockchain can lead to new sustainable business models [44], fostering both financial and social sustainability. Distributed solutions and smart contracts facilitate entrepreneurship and innovation, enabling the creation of new businesses or the development of start-ups. Another feature is transparency, which allows for social proof mechanisms, with people feeling they must behave as others expect. Finally, consensus protocol allows for long-term orientation, with community members having a long-term perspective on fair behavior, trust, and reciprocity.

## 5. Discussion and Conclusions

The objective of this paper was to understand the process underlying the adoption of blockchain technology in accountancy. Indeed, this is an understudied area with respect to innovation and technology that nevertheless could reveal a number of benefits and new business models relevant to organizations and practitioners [10,12,13,15]. In order to understand the perceptions and behaviors of workers, we conducted a meta-synthesis of the qualitative literature on the topic, going to analyze in detail 10 recent studies that put under the magnifying glass the perceived pros and cons when it comes to blockchain technology adoption in accountancy. In sum, we have identified a process that starts from the need to overcome the inefficiencies of a system that is still too little digitized, such as that of accounting, and that leads to new sustainable business models. New developments and innovations related to blockchain technology need to overcome some barriers and resistance to change, also thanks to a number of drivers and positive experiences related to innovation. Although there is a lack of clarity in accountancy regarding the adoption, usage, and applications of blockchain technology [9,11,33], this study recognizes its critical importance and proposes several implications.

### 5.1. Contributions to the Literature

This study contributes to the literature in at least two ways.

First, we address the issue of blockchain in the accountancy domain, which is a topic that has been little addressed in the literature [12,13], especially from an individual perspective. The blockchain provides a number of features and benefits to the accountancy industry and workers, such as: (i) increased efficiency, as well-designed technologies act as fast and powerful databases; (ii) reduced errors, as smart contracts automatically perform many accounting functions once data are in the chain, thereby reducing human error; (iii) facilitated dispute resolution through smart contracts; (iv) fraud reduction, since, in order to change a record, the same change would have to be made simultaneously on all copies of the distributed ledger, which is highly impractical; and (v) reduction in auditing complexity and frequency, due to the automation of many auditing functions.

Second, our findings enable us to offer in-depth knowledge of the process of blockchain technology adoption and to generate useful insights for accountancy, since we used a qualitative approach, which is more varied and broader than a quantitative approach [9]. Using a meta-synthesis of the qualitative literature, our findings highlight the impact of blockchain adoption on the attitudes and behaviors of workers in accountancy. In particular, our findings offer useful insights for academics and practitioners regarding the

main reasons leading organizations to adopt or not adopt, and individuals to use or not use, blockchain technologies in accountancy. Individuals decide to adopt blockchain in accountancy if they can overcome resistance and barriers, and only then can they reap the maximum benefit. The process model we propose starts from the need to challenge and overcome the traditional financial system. A series of scandals have occurred in recent years, mainly due to falsification of records, mutability of information, and liability in organizations, leading scholars and practitioners to identify the blockchain as an antidote to solve inequities and corruption and thus as a motivational push for the adoption of new technologies in the accounting sector. As a second step, when deciding to adopt blockchain in accounting—through the stages of knowledge, persuasion, decision, implementation, and confirmation—one must deal with barriers and drivers. Some barriers to implementation are digital skills; issues of scalability, interoperability, and compatibility; the more general costs of implementing new technologies; and changing regulations and the lack of appropriate governance. Conversely, some drivers include transactional efficiency and disintermediation, data immutability and integrity, and protection against hacking. Unfortunately, resistance to change—mainly due to a lack of knowledge or technological infrastructure—must also be considered in this case. When the drivers are more "convincing" than the barriers, new sustainable accounting practices related to blockchain are adopted, generating new efficient business models and practices based on agile connection and communication between different actors and realities.

These findings open up a variety of opportunities for all companies, including small and medium-sized businesses, which can use new technologies in general, and blockchain in particular, to innovate their business models while seeking sustainability outcomes.

### 5.2. Practical Implications

This study also provides some practical implications. Using blockchain in accountancy means optimizing organizational processes, so practitioners are encouraged to take note of integrated software solutions, set up either directly by the company/professional firm via a Software as a Service (SaaS) approach or with other cloud-based solutions. Without the use of technology, accounting must consider a number of inefficiency-related costs that no organization can sustain in the long run operationally and financially. It is important to know that the use of technology in accountancy has not proven to be an expensive tool to date. There are flexible platforms that accountancy service providers often provide to small businesses for free or with small upfront fees; their use is important because, now more than ever, more work is required to ensure accounts are correct, which means more manual administrative work, taking up more time and human and financial resources. In the course of this transformation, companies that have not begun to adopt technological solutions in their accounting processes will find themselves at a serious disadvantage in two important respects, both of which are directly related to speed: speed in carrying out work and speed in reporting and quality of available detail.

### 5.3. Limitations and Future Research

Despite the theoretical contributions reported, this meta-synthesis has some limitations. First, this methodology may not include papers relevant to the research question, and, despite the methodological rigor with which the articles were selected, it is possible to assume a nonexhaustive inclusion of eligible studies. In addition, the choice not to include studies may omit some interesting insights on the research question. Another limitation is related to the obsolescence of this research, as blockchain is a very current topic and much covered in the literature, and thus it is possible that by the time this article is published, new developments may have emerged on the topic. However, it is certainly important to understand the current state of a phenomenon despite its constant evolution. Blockchain technologies represent an important future development for accounting practices and the accounting profession [39]. At the moment, there is a foundation for this to happen, but more studies are needed on the topic so as to understand the potential risks and

opportunities. Moreover, new and adapted policies that address the nature of accounting standards will need to be developed. Finally, it may be interesting to delve deeper into the topic through qualitative studies so as to report in detail how the people behind the use and exploitation of blockchain technology feel and behave toward it.

**Author Contributions:** Both authors contributed to drafting all sections of the paper. All authors have read and agreed to the published version of the manuscript.

**Funding:** This research received no external funding.

**Institutional Review Board Statement:** Not applicable.

**Informed Consent Statement:** Not applicable.

**Data Availability Statement:** Not applicable.

**Conflicts of Interest:** The authors declare no conflict of interest.

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
