# Peer review of "New Ways of Working through Emerging Technologies: A Meta-Synthesis of the Adoption of Blockchain in the Accountancy Domain"

_jtaer, doi:10.3390/jtaer17020043_

Round 1

Reviewer 1 Report

The strengths of the article is original and interesting considerations with is consistent with the pattern of research. Solid methodology of the research with analysis. Therefore contribution to existing knowledge is considerable. Also advantage of the research is perfect organization & readability.

I cannot find the weaknesses of the assessed article. Model article worthy of imitation.

In generally it is excellent article and very interesting considerations, which is consistent with the pattern of research. A very good review article with the analysis  on the topic under study.

Overall evaluation: article it is suitable for publication in current version.

Author Response

Dear Editor,

We would like to express our gratitude to you and the two anonymous Reviewers for the insightful feedback and suggestions that have helped us improving our manuscript. We have carefully responded to each comment that was raised in the reviews and we do hope that, in doing so, we have been able to strengthen the manuscript’s focus, clarity, and contributions to reach the Journal.

Please note that all original comments are shown in full and that our responses are shown below each comment.

REVIEWER #1

The strengths of the article is original and interesting considerations with is consistent with the pattern of research. Solid methodology of the research with analysis. Therefore contribution to existing knowledge is considerable. Also advantage of the research is perfect organization & readability.

I cannot find the weaknesses of the assessed article. Model article worthy of imitation.

In generally it is excellent article and very interesting considerations, which is consistent with the pattern of research. A very good review article with the analysis on the topic under study.

Overall evaluation: article it is suitable for publication in current version.

Thank you for these comments! We are really happy that you enjoyed the paper.

Reviewer 2 Report

The work is certainly good, but there is a strong discrepancy between the beginning sections (1-2) and the end sections (3-5).

The former are much more confusing than the latter.

The following are some specific suggestions for correcting the paper:

Lines 27-34: Too many times the concept of a node modifying the chain and unclear other things, for example, what is a transaction, what does it do to the ledger? Perhaps a small background section on blockchain that clearly introduces concepts such as smart contracts (line 126), and transactions, would be helpful. 

Line 52: I suggest citing also "Blockchain as IoT Economy Enabler: A Review of Architectural Aspects" (https://doi.org/10.3390/jsan11020020).

It seems to me much more appropriate.

Line 60:  "–few " missing white space

Lines 61-62:   "contributions on" -> "contributions to"

Line 89: "psuedonym" -> "pseudonym"

Lines 106-107:  Why do the authors say "using a sort of consensus mechanism"? it is a consensus mechanism.

Line 132: What is the meaning of BT? Blockchain Technology? If so, please specify it

TABLE 1: missing information and mistake 

In the raw of the article "Crypto-economy and new sustainable business models: Reflections and projections using a case study analysis"; The book title is in the wrong column and missing doi " https://doi.org/10.1002/csr.1954" 

Line 217: "and Discussion sections [34]" possible wrong sentence

Line 224: missing introduction/definition on Cohen’s kappa coefficient

Line 246: The usual preposition to use after 'consequences' is not 'regarding'

Line 474: "This study contributes to the literature in at least three ways." but in the abstract, the authors say "Our findings contribute to the existing literature in at least two ways. "

Author Response

Dear Editor,

We would like to express our gratitude to you and the two anonymous Reviewers for the insightful feedback and suggestions that have helped us improving our manuscript. We have carefully responded to each comment that was raised in the reviews and we do hope that, in doing so, we have been able to strengthen the manuscript’s focus, clarity, and contributions to reach the Journal.

Please note that all original comments are shown in full and that our responses are shown below each comment.

REVIEWER #2

The work is certainly good, but there is a strong discrepancy between the beginning sections (1-2) and the end sections (3-5). The former are much more confusing than the latter.

We hope you will appreciate all the changes made. Thank you for your suggestion!

The following are some specific suggestions for correcting the paper:

Lines 27-34: Too many times the concept of a node modifying the chain and unclear other things, for example, what is a transaction, what does it do to the ledger? Perhaps a small background section on blockchain that clearly introduces concepts such as smart contracts (line 126), and transactions, would be helpful. 

We thank you for this comment; we have tried to improve the introductory part of the paper a little bit, hoping for more clarity now, and we have added a definition to concepts like smart contract that actually was not clearly defined.

Line 52: I suggest citing also "Blockchain as IoT Economy Enabler: A Review of Architectural Aspects" (https://doi.org/10.3390/jsan11020020). It seems to me much more appropriate.

Thank you for this suggestion! It is indeed a very interesting article and in line with our paper; we did not know about it, thank you. We have now added this study in the text and in the reference list.

Line 60:  "–few " missing white space

Thank you!

Lines 61-62:   "contributions on" -> "contributions to"

Thank you! 

Line 89: "psuedonym" -> "pseudonym"

Thank you!

Lines 106-107:  Why do the authors say "using a sort of consensus mechanism"? it is a consensus mechanism.

Thank you for this suggestion, we have corrected.

Line 132: What is the meaning of BT? Blockchain Technology? If so, please specify it

Thank you for this suggestion, in fact you are right: we reported the wording blockchain technology. 

TABLE 1: missing information and mistake 

In the raw of the article "Crypto-economy and new sustainable business models: Reflections and projections using a case study analysis"; The book title is in the wrong column and missing doi " https://doi.org/10.1002/csr.1954" 

Thank you for pointing this out. We have now corrected, putting the publication information in the right column and adding the doi.

Line 217: "and Discussion sections [34]" possible wrong sentence

In fact, it was not a mistake, since you refer to the "Results" and "Discussion" sections. However, we thought it was appropriate to add the quotation marks so that the reader would better understand what we were talking about. 

Line 224: missing introduction/definition on Cohen’s kappa coefficient

Thank you for this suggestion. We have now specified that “this is a statistic used to assess qualitative item inter- and intra-rater reliability and able to consider the potential of agreement occurring by coincidence”.

Line 246: The usual preposition to use after 'consequences' is not 'regarding'

 Thank you for this suggestion, we have now corrected.

Line 474: "This study contributes to the literature in at least three ways." but in the abstract, the authors say "Our findings contribute to the existing literature in at least two ways."

Thank you for this suggestion, because mistakenly in the discussion we split the second contribution related to the process of the adoption of blockchain technology into two. We have now corrected, reporting "at least two" contributions throughout the paper.

Round 2

Reviewer 2 Report

congratulations on a job well done